# Post-earthquake dizziness syndrome following the 2016 Kumamoto earthquakes, Japan

Toru Miwa[1,2]*, Hidetake Matsuyoshi[3], Yasuyuki Nomura[4], Ryosei Minoda[5]

**1** Department of Otolaryngology and Head and Neck Surgery, Tazuke Kofukai Medical Research Institute Kitano Hospital, Osaka, Japan, **2** Department of Otolaryngology and Head and Neck Surgery, Graduate of School of Medicine, Kyoto University, Kyoto, Japan, **3** Matsubase ENT Clinic, Matsubase, Japan, **4** Department of Otolaryngology, Nihon University School of Medicine, Tokyo, Japan, **5** Minoda ENT Clinic/ Kumamoto Ear Clinic, Kumamoto, Japan

* t-miwa@kitano-hp.or.jp

**Data Availability Statement:** All relevant data are within the manuscript and its Supporting Information files.

**Funding:** The authors received no specific funding for this work.

## Abstract

This study aimed to examine the types and causes of dizziness experienced by individuals after a major earthquake. This cross-sectional study enrolled healthy participants who experienced the 2016 Kumamoto earthquakes and their aftershocks. Participants completed a questionnaire survey on their symptoms and experiences after the earthquakes. The primary outcome was the occurrence of dizziness and the secondary outcome was the presence of autonomic dysfunction and anxiety. Among 4,231 eligible participants, 1,543 experienced post-earthquake dizziness. Multivariate logistic regression analysis revealed that age ($\geq$21, $P < .001$), female sex ($P < .001$), floor on which the individual was at the time ($\geq$3, $P = .007$), tinnitus/ear fullness ($P < .001$), anxiety ($P < .001$), symptoms related to autonomic dysfunction ($P = .04$), and prior history of motion sickness ($P = .002$) were significantly associated with the onset of post-earthquake dizziness. Thus suggesting that earthquake-related effects significantly affect inner ear symptoms, autonomic function, and psychological factors. Earthquake-induced disequilibrium may be further influenced by physical stressors, including sensory disruptions induced by earthquake vibrations, changes in living conditions, and autonomic stress. This study increases our understanding of human equilibrium in response to natural disasters.

## Introduction

Major earthquakes are associated with an increased prevalence of psychiatric morbidities [1,2], sleep disorders [3,4], and dizziness [5–10]. The Kumamoto earthquakes on April 14 and 16, 2016 (Fig 1, moment magnitude = 9.0; Shindo = 7) included several high-magnitude vibrations and aftershocks without secondary disasters. Several months after the initial earthquake, significant outbreaks of dizziness were reported over a large area surrounding the earthquake epicenter [8–10].

Although several reports have described post-earthquake dizziness [5–10], its characteristic symptoms remain undefined. After the Tohoku earthquake on March 11, 2011 (moment

**Competing interests:** The authors have declared that no competing interests exist.

magnitude = 9.0; Shindo = 7) and its sequelae (a tsunami and the Fukushima nuclear disaster), Nomura et al. defined the characteristic symptoms of post-earthquake dizziness as *post-earthquake dizziness syndrome* (PEDS) [11]. Healthy individuals (those without vestibular diseases) with PEDS experienced illusory body swaying (lasting <1 minute after the earthquake). Psychological stress and mismatched visual/somatosensory inputs that induce autonomic dysfunction reportedly cause PEDS [11]. Although studies have reported vertigo in some individuals with PEDS after the Tohoku earthquake, the influence of major earthquakes and repetitive aftershocks without a tsunami, as well as the influence of the Fukushima nuclear disaster on post-earthquake dizziness, remains unexamined.

Previous studies on conditions similar to PEDS described its symptoms as "getting one's sea legs" on a multiday sea voyage [12–14] or the wind-induced motion of a tall building [15]. In those studies, researchers stated that the mechanisms for those phenomenon are likely to be

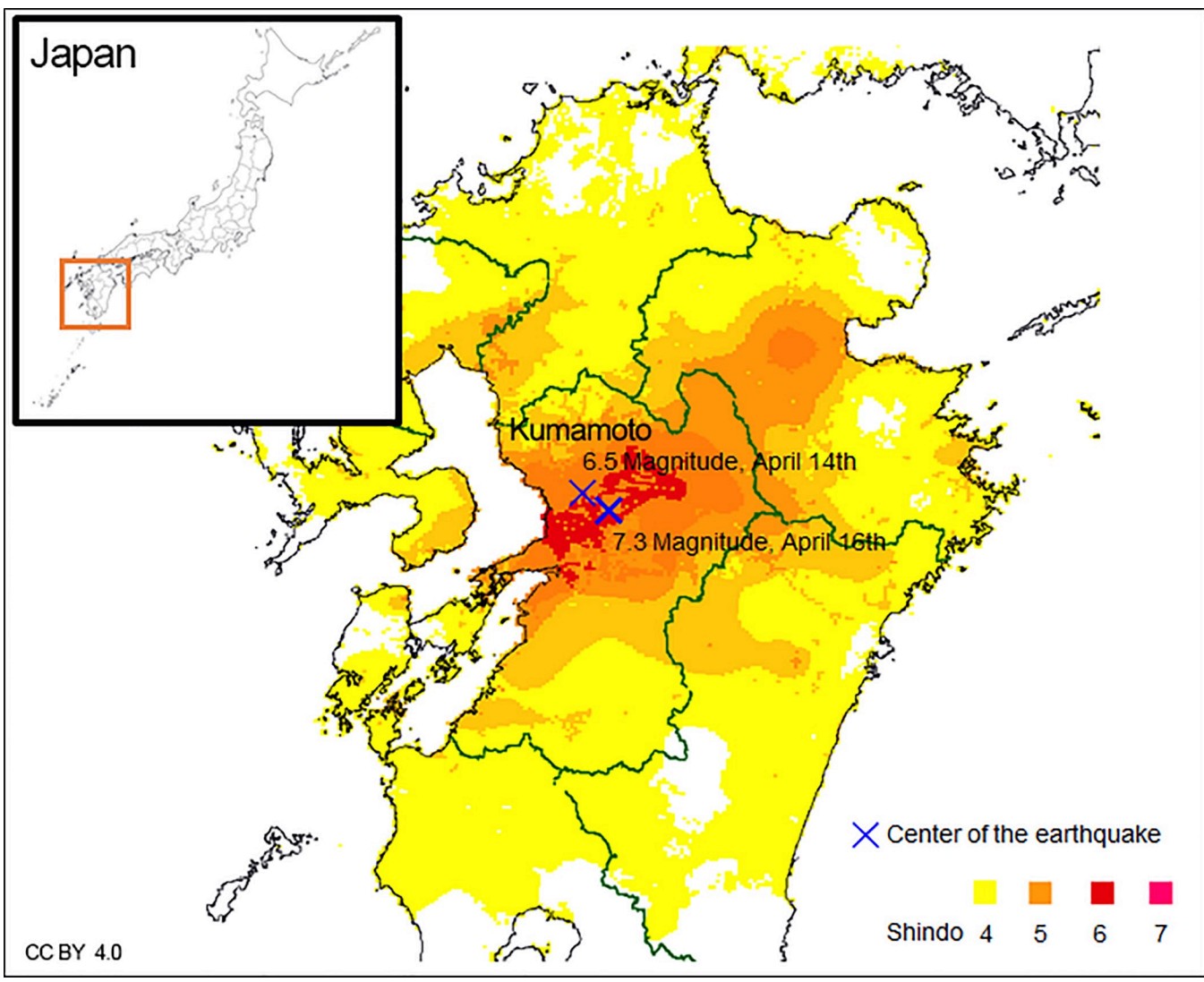

**Fig 1. Map of Kumamoto, 2016.** Blue Xs indicate the epicenters. Magenta to yellow colors indicate earthquake size (Shindo data). Information was reprinted from the Japan Meteorological Agency (https://www.jma.go.jp/jma/indexe.html) under a CC BY license with permission from the Japan Meteorological Agency, original copyright 2016.

explained by the sensory conflict theory/postural instability theory [12–15]. For example, on a ship in the sea, the deck exhibits oscillatory motion simultaneously in six degrees of freedom. The primary power of these oscillations is in the same frequency range (0.1–0.4 Hz) as the oscillations that characterize standing body sway. The onset of motion by the ship leads to a reduction in subjective body stability and balance performance, and individuals tend to exhibit consistent changes in subjective awareness and postural performance [14]. Munafo et al. and Stoffregen et al. concluded that those effects were caused by conscious awareness within the ecological analyses of perception and action, similar to the postural instability theory [12–14]. Although a tall building facing wind typically vibrates in the range of 0.063–1 Hz, random low-frequency building vibration could cause wave interference that is difficult to anticipate. Therefore, an individual counters the natural adjustments that are required to maintain an appropriate posture [15]. Walton et al. concluded that those effects are caused by motion sickness via sensory conflict among visual, vestibular, and proprioceptive sensations [15]. In addition, Stoffregen et al. speculated that the duration of exposure was directly related to the intensity of symptoms experienced, and that the magnitude of movement may also be related to symptom severity [13].

The primary earthquake and repetitive aftershocks caused low frequency (0.1–3.5 Hz) horizontal and vertical linear accelerations. Therefore, the individual was aware of sensory mismatch between visual/proprioceptive sensations and vestibular function, and natural adjustments were required to maintain posture, similar to "getting one's sea legs" [12–14] or the wind-induced motion of a tall building [15].

Thus, this study aimed to examine how major earthquakes and repetitive aftershocks influence human equilibrium. We hypothesized that PEDS after major earthquakes without secondary disasters is caused by a low-dose sensory mismatch, and that after numerous repetitive aftershocks is caused by low-dose postural instability between visual/proprioceptive sensations and vestibular function.

## Materials and methods

### Standard protocol approvals, registrations, and participants' consent

This study adhered to the tenets of the Declaration of Helsinki and was approved by the Kumamoto university institutional review board (Number: 1099). All participants enrolled in the study provided written informed consent.

### Participants

Participants were recruited between June 20 and July 21, 2016, using data collected from a public institution in Kumamoto city. Data from 2017 to 2021 were analyzed. Healthy participants, 10–100 years old, without head trauma who could describe their symptoms and experiences after the Kumamoto earthquakes were included in the study. Individuals who experienced vertigo/dizziness prior to the earthquake were excluded. We included 3,656 healthy individuals who experienced the Kumamoto earthquakes and aftershocks without being exposed to radioactive agents or the tsunamis that hit Kumamoto city for over 12 weeks (Tables 1 and S1). S1 Table shows the patients' demographic information.

### Main measures and outcomes

Participants completed clinical questionnaires and provided information related to age, sex, PEDS symptoms, onset dates, duration of PEDS symptoms, concomitant symptoms, and individual factors. Clinical questionnaires were developed based on those used in the Tohoku

**Table 1. Participants' details.**

| People exposed to Kumamoto earthquake: 3,656 | PEDS group |
|---|---|
| | People who felt the illusion sway: 1,543 |
| | Non-PEDS group |
| | People who did not feel the illusion sway: 2,113 |

PEDS, post-earthquake dizziness syndrome.

earthquake [11] (S2 Table). As geo-structural building conditions may influence the intensity of the shaking experienced [16] and subsequent PEDS, we investigated the building type (wooden or iron), and the floor level on which the participants were at the time of the earthquake, to determine sensory conflict effects. We defined PEDS as the illusion of body swaying; however, participants may consider actual vibrations (aftershocks) as PEDS. Therefore, we investigated correlations between the prevalence of PEDS, geological conditions, and the number of aftershocks in each region. The primary outcome was the occurrence of post-earthquake dizziness, i.e., vertigo/dizziness, experienced by participants after an earthquake. The secondary outcomes were autonomic dysfunction and anxiety.

## Statistical analyses

For the primary outcome, changes in the proportion of patients with PEDS before and after the earthquake were compared using the Chi-squared test. For secondary outcomes, PEDS-associated symptoms were assessed using multivariate logistic regression analysis with age, sex, region, building type, floors in the building, tinnitus/ear fullness, anxiety, symptoms related to autonomic dysfunction for concomitant symptoms, and prior history of motion sickness as the explanatory variables. We also estimated the odds ratios (OR) and 95% confidence intervals (CI). For multivariate logistic regression analysis, the model was created after confirming the variance inflation factor. Imputation was performed for missing values by random forest methods. As a linear relationship was speculated, Pearson's correlation coefficients were used to examine the relationship between the rate of perceived PEDS and number of aftershocks. Power and sample size calculations were conducted before and after data collection using PS software (Ver. 3.1.6, Vanderbilt University, Nashville, TN, USA) [17]. Statistical significance was set at $P < .05$. Evaluations were determined as "not applicable" if the calculated sample size after data collection was found insufficient for statistical analysis. Statistical analyses were performed using GraphPad Prism version 8.0.0 for Windows (GraphPad Software, San Diego, CA, USA, www.graphpad.com).

## Results

### Prevalence of PEDS after the Kumamoto earthquakes

After the Kumamoto earthquake, 1,543 participants (42.2%, 95% CI: 41.7%–46.7%) experienced the illusion of their bodies swaying. Fig 2 (panels a–f) shows the frequency and temporal parameters of PEDS. PEDS was higher in female than in male participants (Fig 2A), and female participants in their 30's were more likely to experience PEDS (73%) than those in other age groups, while male participants in their 40's were more likely to experience PEDS (60%) than those in other age groups (Fig 2B). PEDS was most frequently described as a feeling of the ground shaking, experienced within one week of the earthquake and lasting up to several weeks in an indoor setting (Fig 2C–2F).

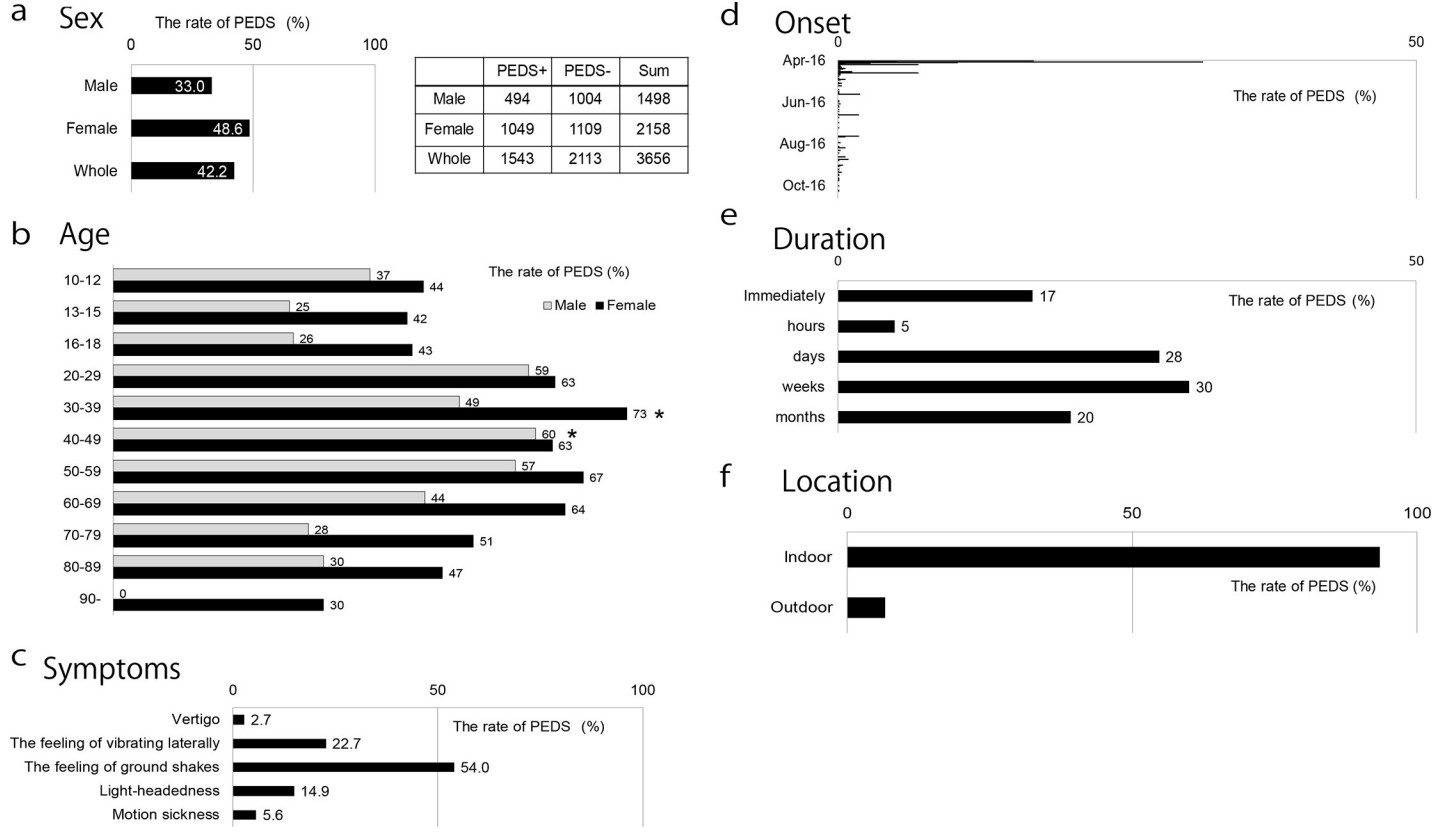

**Fig 2. PEDS frequency according to sex, age, and earthquake parameters.** a) PEDS was present in 1,543 of the 3,656 participants (42.2%). The incidence of PEDS was higher in female than male participants. b) Female participants in their 30's were more likely to experience PEDS (73%, asterisk) than female participants of other ages. Male participants in their 40's were more likely to experience PEDS (60%, asterisk) than male participants of other ages. c) PEDS was most frequently described as a feeling of the ground shaking. d, e) PEDS was felt within one week of the earthquake and lasted up to several weeks. f) PEDS was more commonly experienced in indoor settings. Abbreviation: PEDS, post-earthquake dizziness syndrome.

## Causes of PEDS

Multivariate logistic regression analysis revealed that age ($\geq$ 21 years, OR 3.00, 95% CI: 2.41–3.73, $P <$ .001), female sex (OR 1.71, 95% CI: 1.45–2.01, $P <$ .001), floors in the building ($\geq$ 3, OR 1.35, 95% CI: 1.08–1.67, $P =$ .007), tinnitus/ear fullness (OR 1.43, 95% CI: 1.25–1.64, $P <$ .001), anxiety (OR 3.05, 95% CI: 2.56–3.64, $P <$ .001), symptoms related to autonomic dysfunction (OR 1.10, 95% CI: 1.00–1.21, P = .04), and prior history of motion sickness (OR 1.22, 95% CI: 1.10–1.36, $P =$ .002) were significantly associated with PEDS onset (Table 2). In addition, a propensity analysis adjusted for age and sex revealed that tinnitus/ear fullness (OR 1.48, 95% CI: 1.24–1.79, $P <$ .001), anxiety (OR 3.12, 95% CI: 2.48–3.93, $P <$ .001), symptoms related to autonomic dysfunction (OR 1.30, 95% CI: 0.89–1.43, P = .009), and prior history of motion sickness (OR 1.23, 95% CI: 1.07–1.41, P = .003) were also significant. Thus, PEDS was significantly associated with inner ear symptoms, mood symptoms, and autonomic function. In addition, being on a high floor in a building affected PEDS onset, but this relationship depended on a specific age range and sex. Region (OR 1.07, 95% CI: 0.97–1.18, $P =$ .14) and building type (OR 0.97, 95% CI: 0.81–1.16, $P =$ .74) had no significant association with PEDS.

To exclude the possibility that participants considered actual vibrations (aftershocks) as PEDS, we investigated the correlation between the prevalence of PEDS, geological conditions, and the number of aftershocks in each region. The prevalence of PEDS and geological

**Table 2. Results of the multivariate logistic regression analysis of PEDS onset.**

| Explanatory variables | OR | 95% CI | P-value |
|---|---|---|---|
| (Intercept) | 0.16 | 0.11–0.23 | $P < .001$*** |
| Age ($< 21/\geq 21$ years) | 3.00 | 2.41–3.73 | $P < .001$*** |
| Sex (Male/Female) | 1.71 | 1.45–2.01 | $P < .001$*** |
| Region (South/North/West/Center/East) | 1.07 | 0.97–1.18 | $P = .14$ |
| Building type (Wooden/Iron) | 0.97 | 0.81–1.16 | $P = .74$ |
| Floors of building ($< 3/\geq 3$) | 1.35 | 1.08–1.67 | $P = .007$** |
| Tinnitus/ear fullness | 1.43 | 1.25–1.64 | $P < .001$*** |
| Anxiety | 3.05 | 2.56–3.64 | $P < .001$*** |
| Symptoms related to autonomic dysfunction | 1.10 | 1.00–1.21 | $P = .04$* |
| Prior history of motion sickness | 1.22 | 1.10–1.36 | $P = .002$** |

*$P < .05$

**$P < .01$

***$P < .001$.

Abbreviations: OR, odds ratio; CI: Confidence interval.

conditions or number of aftershocks in each region were not significantly correlated (R (3) = - 0.06, 2.8%, 95% CI: 12.0%–17.2%, $P = .28$; Fig 3; Table 3). While the correlation analysis contained only five data points and is thus limited in statistical power, visual inspection supports the idea that no relationship exists between aftershocks and PEDS.

## Discussion

This study examined the occurrence and possible mechanisms of PEDS after major earthquakes. Our findings revealed a significantly increased prevalence of dizziness after major earthquakes. Specifically, the results from our sample, drawn from the general population,

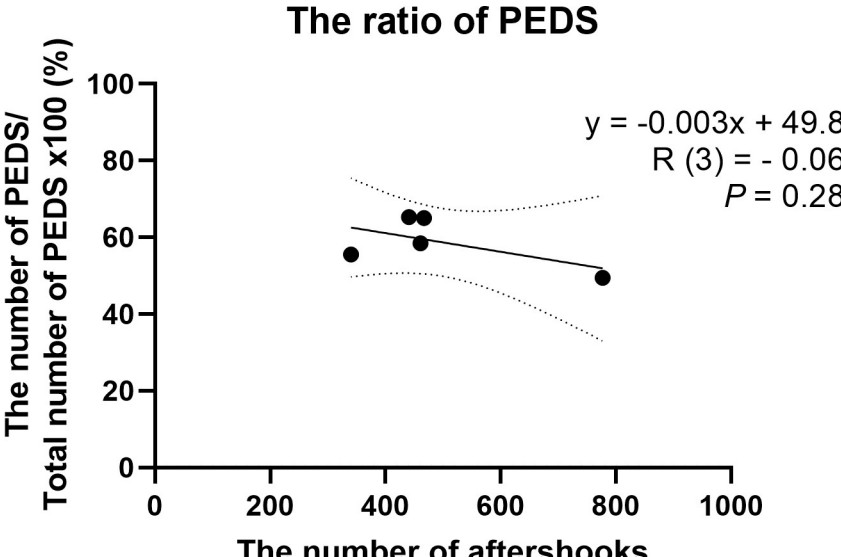

**Fig 3. Relationship between aftershock number and the likelihood of PEDS.** There was a null correlation between geological conditions/number of aftershocks and prevalence of PEDS (R (3) = - 0.06, $P = .28$). Abbreviation: PEDS, post-earthquake dizziness syndrome.

**Table 3. Number of aftershocks and PEDS ratio per region.**

| Kumamoto Region | PEDS+:PEDS- | Max Shindo | Number of aftershocks |
|---|---|---|---|
| Central | 595:1,031 | 7 | 467 |
| Northern | 103:83 | 7 | 340 |
| Southern | 92:66 | 7 | 461 |
| Western | 89:92 | 7 | 777 |
| Eastern | 551:820 | 7 | 441 |

PEDS, post-earthquake dizziness syndrome.

indicate that PEDS occurs after major earthquakes. The prevalence of PEDS in our study was slightly lower than that previously reported [11]. The main outcomes of previous studies [6,11] were influenced by secondary disasters (i.e., tsunami and the Fukushima nuclear disaster), whereas the main outcomes of our study focused exclusively on the effects of major earthquakes. Therefore, since secondary disasters presumably cause mental stress, this may have consequently increased the reported prevalence of PEDS in these studies. Earthquake-induced mental stress may contribute to phobic disorders [18], which may subsequently affect vestibular function and elicit the illusory feeling of swaying. Honma et al. [6] reported a positive correlation between stabilometric parameters and State-Trait Anxiety Inventory scores in individuals who experienced the Tohoku earthquake; however, this correlation was absent in individuals who did not experience the earthquake [6]. The questionnaire survey findings demonstrated that anxiety likely contributed to the observed increase in PEDS prevalence.

To investigate other causes of PEDS after the Kumamoto earthquakes, we examined motion sickness using the sensory conflict theory/postural instability theory as described in the Introduction [12–15,19,20]. The primary earthquake and repetitive aftershocks caused low frequency vibrations (0.1–3.5 Hz), which may cause motion sickness. We observed that age ($\geq 21$), sex (female), and floors in the building ($\geq 3$), as well as visual and somatosensory symptoms such as tinnitus/ear fullness, anxiety, autonomic symptoms (i.e., sweating abnormalities, digestive difficulties, urinary problems, vision problems), and prior history of motion sickness, resulted in an increased incidence of PEDS. Thus, after the Kumamoto earthquake, individuals were aware of sensory conflict between visual/proprioceptive sensations and vestibular function, and natural adjustments or adaptations were required in order to maintain posture. This was similar to the phenomenon of "getting one's sea legs" on a multiday sea voyage [12–14] or wind-induced motion of a tall building [15]. In addition, after the earthquake, individuals experienced numerous repetitive aftershocks for several months. One explanation of our results is that numerous repetitive aftershocks caused low-level motion sickness via the postural instability theory [13] and sensory conflict theory [15].

Additionally, our findings demonstrate that the prevalence of PEDS did not directly correspond to the actual vibrations (i.e., repetitive aftershocks). The epicenter of the Kumamoto earthquakes was onshore, whereas that of the Tohoku earthquakes was offshore. Generally, normal faults (vertical direction) cause onshore earthquakes, whereas strike-slip and normal faults (vertical and horizontal directions) cause offshore earthquakes [21]. Thus, the mechanical relationship between vibration types and the development of PEDS after the Kumamoto earthquakes may have differed from that after the Tohoku earthquakes [6]. These findings and those of previous studies suggest that a higher-order information mismatch between the visual or proprioceptive systems and vestibular systems caused by a major earthquake can produce spatial disorientation, dizziness, and more pronounced autonomic symptoms [19,20]. It is

similar to somatic sensations, such as those caused by startling stimuli, and often induces an autonomic reflex, termed as the *startle response* [22] or phobic postural vertigo [23].

In summary, we speculate that an earthquake-induced disequilibrium is further influenced by physical stressors, including sensory disruptions instigated by earthquake vibrations, changes in living conditions, and autonomic stress.

This study was limited by the fact that psychological stress was not fully investigated. Investigating the impact of post-disaster psychological stress on healthy individuals with PEDS would help verify the causes of post-earthquake dizziness. In addition, we cannot make any strong statements about the mechanistic cause mediating our results (sensory conflict/postural stability) as our study was not designed to test any one theory.

## Conclusions

Exposure to major earthquakes and aftershocks induced post-earthquake dizziness in a significant percentage of individuals. Our results indicate that post-earthquake dizziness may be due to sensory conflicts/postural instability mediated by vestibular dysfunction, autonomic dysfunction, and/or psychological factors. Our findings can facilitate the management of dizziness experienced during or after disasters. Future studies should identify ways to mitigate autonomic dysfunction and prevent post-earthquake dizziness.

## Supporting information

**S1 Table. Patients' demographic information.**
(DOCX)

**S2 Table. PEDS questionnaire.**
(DOCX)

## Acknowledgments

We would like to thank the volunteers and school teachers for their cooperation with the completion of the questionnaire survey. We thank Saki Miwa for helping with data processing during various stages of this research project. We also thank Editage (www.editage.jp) for English language editing and publication support.

## Author Contributions

**Conceptualization:** Toru Miwa.

**Data curation:** Toru Miwa.

**Formal analysis:** Toru Miwa.

**Investigation:** Toru Miwa.

**Methodology:** Toru Miwa, Hidetake Matsuyoshi, Yasuyuki Nomura, Ryosei Minoda.

**Project administration:** Toru Miwa, Hidetake Matsuyoshi, Yasuyuki Nomura, Ryosei Minoda.

**Resources:** Toru Miwa.

**Software:** Toru Miwa.

**Supervision:** Toru Miwa, Hidetake Matsuyoshi, Yasuyuki Nomura, Ryosei Minoda.

**Validation:** Toru Miwa.

**Visualization:** Toru Miwa.

**Writing – original draft:** Toru Miwa.

**Writing – review & editing:** Toru Miwa.

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
