## [Decision Letter · Decision Letter 0]

27 May 2021

PONE-D-21-13080

Post-earthquake dizziness syndrome following the 2016 Kumamoto earthquakes, Japan

PLOS ONE

Dear Dr. Miwa,

Thank you for submitting your manuscript to PLOS ONE. After careful consideration, we feel that it has merit but does not fully meet PLOS ONE’s publication criteria as it currently stands. Therefore, we invite you to submit a revised version of the manuscript that addresses the points raised during the review process.

This is a fascinating study, and I applaud your initiative in creating it under what must have been exceptionally challenging conditions. Two highly qualified Reviewers have offered their comments. In revising, please attend carefully to the comments from both Reviewers. In addition, as your study relates phenomenology to movement of the support surface, it might be helpful for you to consider one of my own articles:

Munafo, J., Wade, M. G., Stergiou, N., & Stoffregen, T. A. (2015). Subjective reports and postural performance among older adult passengers on a sea voyage. *Ecological Psychology*, *27*, 127-143.

We look forward to receiving your revised manuscript.

Kind regards,

Thomas A Stoffregen, PhD

Academic Editor

PLOS ONE

Journal Requirements:

2. We note that Figure 1 in your submission contains map images which may be copyrighted.

We require you to either (a) present written permission from the copyright holder to publish this figure specifically under the CC BY 4.0 license, or (b) remove the figure from your submission:

b. If you are unable to obtain permission from the original copyright holder to publish this figure under the CC BY 4.0 license or if the copyright holder’s requirements are incompatible with the CC BY 4.0 license, please either i) remove the figure or ii) supply a replacement figure that complies with the CC BY 4.0 license. Please check copyright information on all replacement figures and update the figure caption with source information. If applicable, please specify in the figure caption text when a figure is similar but not identical to the original image and is therefore for illustrative purposes only.

Reviewers' comments:

Reviewer's Responses to Questions

**Comments to the Author**

1. Is the manuscript technically sound, and do the data support the conclusions?

Reviewer #1: Yes

Reviewer #2: Partly

2. Has the statistical analysis been performed appropriately and rigorously? 

Reviewer #1: I Don't Know

Reviewer #2: Yes

3. Have the authors made all data underlying the findings in their manuscript fully available?

Reviewer #1: No

Reviewer #2: Yes

4. Is the manuscript presented in an intelligible fashion and written in standard English?

Reviewer #1: Yes

Reviewer #2: Yes

5. Review Comments to the Author

Reviewer #1: This paper outlines a cross-sectional analysis of post-earthquake dizziness syndrome (PEDS) following the 2016 series of tremors in Kumamoto, Japan. Results showed that more than one third of participants reported post-earthquake dizziness. Several factors were identified as related to PEDS, including female sex, ear fullness, an age of 21 or more, and anxiety. The authors conclude that earthquakes affect inner-ear function, autonomic symptoms, and psychological factors, both directly and indirectly via changes in environmental factors.

The paper offers a useful improvement to our knowledge about post-earthquake symptoms. The principal concern is that I would like to see a greater explanation of the mechanistic link between tremors and the outcomes measured here. A possible error in reporting statistics emerged (this is potentially a minor issue). Several other improvements could be made, as described below.

The Discussion is brief and leaves the reader with some lingering questions. As in the Introduction, the hypothesis linking earthquakes with lingering dizziness surely needs to be discussed at greater length. The statement by the authors “we examined sensory conflict theory and motion sickness” is true only to a very limited extent, this could be significantly expanded. I advise the authors to consider articles on postural stability and motion sickness (e.g. Stoffregen et al. 2013; Walton et al., 2011) as there seems to be a natural link between this research and the current study.

Could the authors please check that they are correctly stating Pearson r values for their correlations? A negative correlation should be associated with a negative r value, but L137 states a positive value (note, degrees of freedom should also be added).

The authors asked participants to report “Autonomic symptoms” in their questionnaire. It must be clarified in the text what was specifically meant by “autonomic symptoms” and how they were described to participants.

It seems appropriate to reference the findings of a recent overview paper with clinical patients conducted by the main author (Miwa, 2021).

L23 delete “the”

L34 I would press the authors to say more about their hypothesis. They state a multisensory mismatch produces PEDS, but this is descriptive and not explanatory. How could a single-shot mismatch result in PEDS, or are aftershocks likely to be a necessary factor? Are there existing models that support this hypothesis?

L50 It would be valuable to state that these participants reported no head trauma brought about by the earthquake. I am assuming that those data were obtained. If they were not collected, this could be added as a limitation.

L69 “changes in the proportion of patients with PEDS … were compared by hypothesis testing” This does not specify the type of test used (e.g. t-tests), which should be included here.

L74 Could the authors specify what proportion of the responses were imputed?

L87 The number of participants in this study should be stated in the ‘Participants’ section.

L122 The authors could briefly state here what the non-significant factors were (e.g. “Building type and regional location had no significant association with PEDS”).

L133 A caveat here is that the correlation contains very few datapoints, and this should be stated by the authors in text, not a figure caption (e.g. “While the correlation analysis contained only 5 datapoints and is thus limited in statistical power, visual inspection supports the idea that no relationship exists between aftershocks and PEDS”).

In addition, Table 3 references “earthquake sickness ratio”, I assume this should be corrected to PEDS. Column 1 in Table 3 can also be removed, with ‘Kumamoto Region’ being the heading of column 2.

L135 “Despite the negative correlation between geological conditions/number of aftershocks and prevalence of PEDS” I would call this a ‘null’ correlation, not a negative one.

I advise adding significance asterisks in (e.g.) Fig 2b.

References

Miwa, T. (2020). Vestibular function after the 2016 Kumamoto earthquakes: a retrospective chart review. Frontiers in Neurology, 11.

Stoffregen, T. A., Chen, F. C., Varlet, M., Alcantara, C., & Bardy, B. G. (2013). Getting your sea legs. PLoS One, 8(6), e66949.

Walton, D., Lamb, S., & Kwok, K. C. (2011). A review of two theories of motion sickness and their implications for tall building motion sway. Wind and Structures, 14(6), 499.

Reviewer #2: The examination of post-earthquake effects on perception of balance and movement is an interesting phenomenon that the authors were able to take advantage of after the Kumamoto earthquake. The data collection and analysis were well done. However, it is not clear why the authors are couching this a sensory mismatch issue. That is why is this the theoretical mechanism that is thought to account to the phenomenological data they collected. There needs to be a clear justification of why mismatch perspectives are the right model to use. Without that connection there isn't a strong justification for the research questions. Are there other alternatives that could also account for your findings?

6. PLOS authors have the option to publish the peer review history of their article (what does this mean?). If published, this will include your full peer review and any attached files.

Reviewer #1: **Yes: **Séamas Weech

Reviewer #2: **Yes: **L. James Smart Jr.

---

## [Author Response · Author response to Decision Letter 0]

14 Jun 2021

Reviewer #1: This paper outlines a cross-sectional analysis of post-earthquake dizziness syndrome (PEDS) following the 2016 series of tremors in Kumamoto, Japan. Results showed that more than one third of participants reported post-earthquake dizziness. Several factors were identified as related to PEDS, including female sex, ear fullness, an age of 21 or more, and anxiety. The authors conclude that earthquakes affect inner-ear function, autonomic symptoms, and psychological factors, both directly and indirectly via changes in environmental factors.

The paper offers a useful improvement to our knowledge about post-earthquake symptoms. The principal concern is that I would like to see a greater explanation of the mechanistic link between tremors and the outcomes measured here. A possible error in reporting statistics emerged (this is potentially a minor issue). Several other improvements could be made, as described below.

The Discussion is brief and leaves the reader with some lingering questions. As in the Introduction, the hypothesis linking earthquakes with lingering dizziness surely needs to be discussed at greater length. The statement by the authors “we examined sensory conflict theory and motion sickness” is true only to a very limited extent, this could be significantly expanded. I advise the authors to consider articles on postural stability and motion sickness (e.g. Stoffregen et al. 2013; Walton et al., 2011) as there seems to be a natural link between this research and the current study.

Response: I appreciate your valuable comments. I have added the description of motion sickness by postural stability theory in the Discussion.

Could the authors please check that they are correctly stating Pearson r values for their correlations? A negative correlation should be associated with a negative r value, but L137 states a positive value (note, degrees of freedom should also be added).

Response: Thank you for your valuable comment. I have corrected it and added degrees of freedom.

The authors asked participants to report “Autonomic symptoms” in their questionnaire. It must be clarified in the text what was specifically meant by “autonomic symptoms” and how they were described to participants.

Response: Thank you for your kind advice. I have added specific examples in the main text and Table S2.

It seems appropriate to reference the findings of a recent overview paper with clinical patients conducted by the main author (Miwa, 2021).

Response: Thank you for the suggestion. I have added the information accordingly.

L23 delete “the”

Response: Thank you for the suggestion. I have deleted it accordingly.

L34 I would press the authors to say more about their hypothesis. They state a multisensory mismatch produces PEDS, but this is descriptive and not explanatory. How could a single-shot mismatch result in PEDS, or are aftershocks likely to be a necessary factor? Are there existing models that support this hypothesis?

Response: I appreciate your comment. I have added the required information to the Introduction.

L50 It would be valuable to state that these participants reported no head trauma brought about by the earthquake. I am assuming that those data were obtained. If they were not collected, this could be added as a limitation.

Response: Thank you. I have added the description regarding this in the Materials and Methods section.

L69 “changes in the proportion of patients with PEDS … were compared by hypothesis testing” This does not specify the type of test used (e.g. t-tests), which should be included here.

Response: Thank you for your comment. I have corrected it accordingly.

L74 Could the authors specify what proportion of the responses were imputed?

Response: Yes. The values of building type were imputed because some building occupants did not know to live in ironed-type buildings or wooden-type buildings.

L87 The number of participants in this study should be stated in the ‘Participants’ section.

Response: Thank you for your comment. I have stated the number of participants in the ‘Participants’ section.

L122 The authors could briefly state here what the non-significant factors were (e.g. “Building type and regional location had no significant association with PEDS”).

Response: Thank you for your comment. I have added the required information to the revised manuscript.

L133 A caveat here is that the correlation contains very few datapoints, and this should be stated by the authors in text, not a figure caption (e.g. “While the correlation analysis contained only 5 datapoints and is thus limited in statistical power, visual inspection supports the idea that no relationship exists between aftershocks and PEDS”).

In addition, Table 3 references “earthquake sickness ratio”, I assume this should be corrected to PEDS. Column 1 in Table 3 can also be removed, with ‘Kumamoto Region’ being the heading of column 2.

Response: Thank you for your valuable comments. I have added the suggested description in the main text. In addition, I have corrected “earthquake sickness ratio” to “PEDS”, following your suggestion. In Table 3, column 1 was removed, and I added ‘Kumamoto region’ in the heading of column 2.

L135 “Despite the negative correlation between geological conditions/number of aftershocks and prevalence of PEDS” I would call this a ‘null’ correlation, not a negative one.

Response: Thank you. I have corrected accordingly.

I advise adding significance asterisks in (e.g.) Fig 2b.

Response: Thank you for the valuable advice. According to your suggestion, I have added significance asterisks in Fig 2b.

References

Miwa, T. (2020). Vestibular function after the 2016 Kumamoto earthquakes: a retrospective chart review. Frontiers in Neurology, 11.

Stoffregen, T. A., Chen, F. C., Varlet, M., Alcantara, C., & Bardy, B. G. (2013). Getting your sea legs. PLoS One, 8(6), e66949.

Walton, D., Lamb, S., & Kwok, K. C. (2011). A review of two theories of motion sickness and their implications for tall building motion sway. Wind and Structures, 14(6), 499. 

Munafo, J., Wade, M. G., Stergiou, N., & Stoffregen, T. A. (2015). Subjective reports and postural performance among older adult passengers on a sea voyage. Ecological Psychology, 27, 127-143.

Reviewer #2: The examination of post-earthquake effects on perception of balance and movement is an interesting phenomenon that the authors were able to take advantage of after the Kumamoto earthquake. The data collection and analysis were well done. However, it is not clear why the authors are couching this a sensory mismatch issue. That is why is this the theoretical mechanism that is thought to account to the phenomenological data they collected. There needs to be a clear justification of why mismatch perspectives are the right model to use. Without that connection there isn't a strong justification for the research questions. Are there other alternatives that could also account for your findings?

Response: Thank you for your valuable comments. I have expanded the Discussion section and reconsidered adding the theory on motion sickness (i.e. postural instability theory and sensory conflict theory cited from Stoffregen et al. 2013; Walton et al., 2011). In this study, I speculated that both theories were associated with PEDS. I have added them in the Discussion section. (Lines 181-191)

---

## [Decision Letter · Decision Letter 1]

5 Jul 2021

PONE-D-21-13080R1

Post-earthquake dizziness syndrome following the 2016 Kumamoto earthquakes, Japan

PLOS ONE

Dear Dr. Miwa,

Thank you for submitting your manuscript to PLOS ONE. After careful consideration, we feel that it has merit but does not fully meet PLOS ONE’s publication criteria as it currently stands. Therefore, we invite you to submit a revised version of the manuscript that addresses the points raised during the review process.

Both Reviewers are pleased with your changes, but each asks for some additional clarification. I agree with Reviewer 2 that the sensory conflict theory of motion sickness is not compatible with the postural instability theory. They are mutually exclusive; so much so that the postural instability theory makes an explicit claim that sensory conflict does not even exist.

We look forward to receiving your revised manuscript.

Kind regards,

Thomas A Stoffregen, PhD

Academic Editor

PLOS ONE

Journal Requirements:

Reviewers' comments:

Reviewer's Responses to Questions

**Comments to the Author**

1. If the authors have adequately addressed your comments raised in a previous round of review and you feel that this manuscript is now acceptable for publication, you may indicate that here to bypass the “Comments to the Author” section, enter your conflict of interest statement in the “Confidential to Editor” section, and submit your "Accept" recommendation.

Reviewer #1: (No Response)

Reviewer #2: (No Response)

2. Is the manuscript technically sound, and do the data support the conclusions?

Reviewer #1: Yes

Reviewer #2: Yes

3. Has the statistical analysis been performed appropriately and rigorously? 

Reviewer #1: Yes

Reviewer #2: Yes

4. Have the authors made all data underlying the findings in their manuscript fully available?

Reviewer #1: No

Reviewer #2: Yes

5. Is the manuscript presented in an intelligible fashion and written in standard English?

Reviewer #1: No

Reviewer #2: Yes

6. Review Comments to the Author

Reviewer #1: I am confused as to why the authors include only one to two sentences on the mechanism linking earthquake with sickness (i.e. L120-126). Please be aware, many readers will not be familiar with the idea that sensory conflict or postural instability are thought to cause motion sickness. As a result, it is worthwhile to take the time and explain those ideas in some more detail here. Not too much--this is clearly not the focus of the paper--but I would expect more than the manuscript currently contains.

Indeed, the Discussion contains some of this relevant information that belongs in the Introduction. L273-297 describes previous literature and theory details that I would expect to read before the hypothesis is stated. Please consider re-phrasing this information and transposing it to the Introduction. It can then be referred to in the Discussion: How do your results fit with this literature?

Related to this point, the authors' new inclusions regarding sensory conflict theory and motion sickness were not clear. For example, "In addition, numerous repetitive aftershocks caused low dosage motion sickness via the postural instability theory/ecological approach to perception and action." (L317). There is insufficient evidence for the authors to make this claim as it is written, without qualifiers ("One explanation of our results is that..."). It is also unclear what is the 'ecological approach to perception and action' to a non-expert reader, so it seems appropriate to delete those last 6 words.

It is ultimately true that the authors cannot make any strong statements about what was the mechanistic cause for their results (sensory conflict/postural stability), as their study was not designed to test any one theory. So, they should make this point clear in their limitations. Please note, it is not a strong limitation: This is a very interesting paper, even if it cannot speak to the mechanisms at play in motion sickness.

Remove 'this correlation was not significant' (L248).

L300-301: "motion sickness": Perhaps this should this state "prior history of motion sickness"?

The authors should review the the newly added material as there are several language mistakes.

Reviewer #2: You have thoughtfully addressed the initial issues raised by the reviewers - the intent of my particular comments was not to make you change your theory argument, just justify it. However, I think your revisions on this issue are reasonable. My only suggestion is to make sure to be clear that postural/ecological perspectives on motion sickness do not support sensory conflict explanations. Phenomenologically, the participants experiences might suggest conflict (which is where I think you were going originally) - but as you noted, definitive claims cannot be made at this point.

7. PLOS authors have the option to publish the peer review history of their article (what does this mean?). If published, this will include your full peer review and any attached files.

Reviewer #1: **Yes: **Seamas Weech

Reviewer #2: **Yes: **L. James Smart Jr.

---

## [Author Response · Author response to Decision Letter 1]

22 Jul 2021

Reviewer #1: I am confused as to why the authors include only one to two sentences on the mechanism linking earthquake with sickness (i.e. L120-126).

Please be aware, many readers will not be familiar with the idea that sensory conflict or postural instability are thought to cause motion sickness. As a result, it is worthwhile to take the time and explain those ideas in some more detail here. Not too much--this is clearly not the focus of the paper--but I would expect more than the manuscript currently contains.

Response: Thank you for your valuable comments. I have made sure that in the revision, these concepts are explained with enough detail that readers will be sufficiently informed. 

Indeed, the Discussion contains some of this relevant information that belongs in the Introduction. L273-297 describes previous literature and theory details that I would expect to read before the hypothesis is stated. Please consider re-phrasing this information and transposing it to the Introduction. It can then be referred to in the Discussion: How do your results fit with this literature?

Response: We appreciate your suggestion and I have rephrased the literature information and transposed it into the Introduction. In addition, I made sure to point them out and integrate them into our results in the discussion section. I have considered that stimulation via environment, such as an earthquake, a ship, a building facing wind, causes various degrees of motion sickness. 

Related to this point, the authors' new inclusions regarding sensory conflict theory and motion sickness were not clear. For example, "In addition, numerous repetitive aftershocks caused low dosage motion sickness via the postural instability theory/ecological approach to perception and action." (L317). There is insufficient evidence for the authors to make this claim as it is written, without qualifiers ("One explanation of our results is that..."). It is also unclear what is the 'ecological approach to perception and action' to a non-expert reader, so it seems appropriate to delete those last 6 words.

Response: Thank you for your comments. I agree with the assessment made by the reviewer and have deleted the 'ecological approach to perception and action'. Further, I have added the qualifier "One explanation of our results is that..." in this sentence. My intention was to get across that the mechanisms of motion sickness were different whether it was instigated by the major earthquake or the repetitive aftershocks.

It is ultimately true that the authors cannot make any strong statements about what was the mechanistic cause for their results (sensory conflict/postural stability), as their study was not designed to test any one theory. So, they should make this point clear in their limitations. Please note, it is not a strong limitation: This is a very interesting paper, even if it cannot speak to the mechanisms at play in motion sickness.

Response: I greatly appreciate your kind advice. I have added this concern to the limitation section.

Remove 'this correlation was not significant' (L248).

Response: Thank you for the advice, I have removed this statement.

L300-301: "motion sickness":Perhaps this should this state "prior history of motion sickness"?

Response: Your suggested wording does better reflect our intended meaning. I have revised this statement throughout the manuscript.

The authors should review the the newly added material as there are several language mistakes.

Response: I apologize for this oversight. We have edited the manuscript to ensure it meets adequate English language standards. 

Reviewer #2: You have thoughtfully addressed the initial issues raised by the reviewers - the intent of my particular comments was not to make you change your theory argument, just justify it. However, I think your revisions on this issue are reasonable. My only suggestion is to make sure to be clear that postural/ecological perspectives on motion sickness do not support sensory conflict explanations. Phenomenologically, the participants experiences might suggest conflict(which is where I think you were going originally) - but as you noted, definitive claims cannot be made at this point.

Response: Thank you for valuable comments. After your suggestion, I have reconsidered the mechanisms underlying earthquake-related dizziness, and concluded that the major earthquake and the repetitive aftershocks caused motion sickness via different mechanisms. I have described this in the manuscript. However, as you noted, the stated mechanism is just speculation. As such, I have also mentioned this in the limitation section.

---

## [Editor Report · Decision Letter 2]

26 Jul 2021

Post-earthquake dizziness syndrome following the 2016 Kumamoto earthquakes, Japan

PONE-D-21-13080R2

Dear Dr. Miwa,

We’re pleased to inform you that your manuscript has been judged scientifically suitable for publication and will be formally accepted for publication once it meets all outstanding technical requirements.

Kind regards,

Thomas A Stoffregen, PhD

Academic Editor

PLOS ONE
---

## [Editor Report · Acceptance letter]

28 Jul 2021

PONE-D-21-13080R2 

Post-earthquake dizziness syndrome following the 2016 Kumamoto earthquakes, Japan 

Dear Dr. Miwa:

I'm pleased to inform you that your manuscript has been deemed suitable for publication in PLOS ONE. Congratulations! Your manuscript is now with our production department. 

Kind regards, 

on behalf of

Dr. Thomas A Stoffregen 

Academic Editor

PLOS ONE